# Modulation of Cisplatin Sensitivity through TRPML1-Mediated Lysosomal Exocytosis in Ovarian Cancer Cells: A Comprehensive Metabolomic Approach

**DOI:** 10.3390/cells13020115

**Published:** 2024-01-08

**Authors:** Boyun Kim, Gaeun Kim, Heeyeon Kim, Yong Sang Song, Jewon Jung

**Affiliations:** 1Department of SmartBio, College of Life and Health Science, Kyungsung University, Busan 48434, Republic of Korea; boyunism@gmail.com (B.K.); kge1406@gmail.com (G.K.); 2WCU Biomodulation, Department of Agricultural Biotechnology, Seoul National University, Seoul 08826, Republic of Korea; heekim312@snu.ac.kr (H.K.); yssong@snu.ac.kr (Y.S.S.); 3Cancer Research Institute, College of Medicine, Seoul National University, Seoul 03080, Republic of Korea; 4Department of Obstetrics and Gynecology, College of Medicine, Seoul National University, Seoul 03080, Republic of Korea

**Keywords:** TRPML1, lysosome, exocytosis, chemoresistance, metabolomics, cisplatin, arginine, ovarian cancer

## Abstract

Background: The lysosome has emerged as a promising target for overcoming chemoresistance, owing to its role in facilitating the lysosomal sequestration of drugs. The lysosomal calcium channel TRPML1 not only influences lysosomal biogenesis but also coordinates both endocytosis and exocytosis. This study explored the modulation of cisplatin sensitivity by regulating TRPML1-mediated lysosomal exocytosis and identified the metabolomic profile altered by TRPML1 inhibition. Methods: We used four types of ovarian cancer cells: two cancer cell lines (OVCAR8 and TOV21G) and two patient-derived ovarian cancer cells. Metabolomic analyses were conducted to identify altered metabolites by TRPML1 inhibition. Results: Lysosomal exocytosis in response to cisplatin was observed in resistant cancer cells, whereas the phenomenon was absent in sensitive cancer cells. Through the pharmacological intervention of TRPML1, lysosomal exocytosis was interrupted, leading to the sensitization of resistant cancer cells to cisplatin treatment. To assess the impact of lysosomal exocytosis on chemoresistance, we conducted an untargeted metabolomic analysis on cisplatin-resistant ovarian cancer cells with TRPML1 inhibition. Among the 1446 differentially identified metabolites, we focused on 84 significant metabolites. Metabolite set analysis revealed their involvement in diverse pathways. Conclusions: These findings collectively have the potential to enhance our understanding of the interplay between lysosomal exocytosis and chemoresistance, providing valuable insights for the development of innovative therapeutic strategies.

## 1. Introduction

Ovarian cancer, with its high lethality among gynecological cancers, globally ranked as the third most common in 2020, constituting 5% of female cancer-related deaths [1]. Ovarian cancer is frequently diagnosed at an advanced stage, resulting in a 5-year survival rate of only 17% for patients [2,3]. This attributes to challenges in early detection and development of chemoresistance [4]. Combination chemotherapy involving platinum-based drugs and paclitaxel represents the standard treatment for ovarian cancer [5]. While the majority of ovarian cancer patients initially respond to this standard treatment, recurrence occurs in up to 80% of patients, primarily due to the development of platinum-resistance [6]. Chemoresistance to the standard therapy poses a critical barrier in the treatment of ovarian cancer, resulting in a decrease in the 5-year survival rate [7,8]. Multiple mechanisms contribute to the development of resistance to chemotherapeutics, including tumor heterogeneity, microenvironmental effects, and interruptions of drug access to the target compartment [6]. Unraveling strategies to overcome chemoresistance is imperative for enhancing the survival rate of ovarian cancer, especially in the absence of an optimal biomarker for early detection. Addressing these challenges will be pivotal in advancing the treatment landscape for ovarian cancer and improving patient outcomes.

Lysosomes have mainly been considered membrane-bound vesicles responsible for the degradation of extracellular materials through endocytosis, the recycling of intracellular substances via phagocytosis, and the extrusion of materials from the cell through exocytosis [9,10]. Recently, accumulating evidence has underscored the critical role of lysosomes which have emerged as a significant obstacle in drug access to the target organelles. The acidity of lysosomes creates an environment conducive to the accumulation of chemotherapeutic drugs with weak bases, resulting in their sequestration within these organelles [11,12]. This sequestration acts as a protective mechanism, preventing potential lysosomal damage. However, it also hinders drug access to the target compartment, leading to a clearance process mediated by lysosomal exocytosis [9,12]. This clearance process, in turn, contributes to the development of chemoresistance. Considering this challenge, emerging evidence points to a novel aspect of lysosomal function. Lysosomes can initiate calcium signaling through the TRPML1/TFEB pathway, facilitating lysosomal exocytosis and the clearance of accumulated materials [13,14]. TRPML1, a major calcium-permeable channel present in the lysosomal membrane, plays a key role in the fusion of endolysosomes with the plasma membrane through calcium regulation and serves as a nutrient sensor in cancers [15,16,17,18]. Additionally, the high expression of TRPML1 is observed in various tumor types, including head and neck cancer, pancreatic ductal adenocarcinoma, melanoma, and endometrial cancer. This heightened expression impacts the proliferation and migration of cancer cells, contributing to poor prognosis and low survival rates [19,20,21,22]. Given this collective evidence, exploring interventions that manipulate lysosomal exocytosis, particularly through targeting TRPML1, could be an innovative approach to overcoming chemoresistance in cancer treatment. 

Alterations in cellular metabolism play a crucial role in promoting tumor development and progression, and the metabolic reprogramming of tumors contributes to drug resistance in chemotherapeutics [23]. Global analysis of metabolites (metabolomics) in biological specimens such as urine, blood, and tumors has been utilized to discriminate between non-tumor and tumor patients [24,25,26,27,28]. This approach holds the potential for identifying biomarkers for early diagnosis and prognosis. Moreover, the metabolomic profile of drug-resistant tumors can offer insights into potential mechanisms of chemoresistance. Distinctive metabolomic profiles have been reported in ovarian cancer cells, potentially contributing to chemoresistance. These variations include differences in cysteine and methionine metabolism, arginine and ornithine metabolism, etc., between platinum-sensitive A2780 and platinum-resistant C200 cells [29]. In this study, we conducted a metabolomic analysis to identify specific metabolites altered in response to an inhibitor of the lysosomal calcium channel TRPML1, aiming to enhance the sensitivity of chemoresistant ovarian cancer cells to cisplatin. 

## 2. Materials and Methods

### 2.1. Cell Culture

Ovarian cancer cell lines (TOV21G and OVCAR8) obtained from the American Type Culture Collection (ATCC) and HK2 cells obtained from the Korean Cell Line Bank (KCLB) were cultured in RPMI-1640 (Welgene, Gyeongsan, Republic of Korea) supplemented with 4.5 g/L D-glucose, 2 mM L-glutamine, 10 mM HEPES, 1 mM sodium pyruvate, 1.5 g/L sodium bicarbonate, 10% FBS (Thermo Fisher Scientific, Waltham, MA, USA), 100 U/mL penicillin, and 100 g/mL streptomycin (Thermo Fisher Scientific). A8 and A39 cells were established from the ascites isolated from ovarian cancer patients (supported by Dr. Yong Sang Song at Seoul National University Hospital). The cultured cells were incubated for three days for maintenance at 37 °C in a humidified atmosphere with 5% CO_2_. All cancer cells for further analyses were seeded at a density of 1.3–2.0 × 10^4^ cells/cm^2^.

### 2.2. Information on Patient-Derived Ovarian Cancer Cell Lines

The A8 and A39 cell lines were established from ascites derived from two patients with serous ovarian cancer at stage IIIC. This study received approval from the Institutional Review Board (IRB) at Seoul National University Hospital (Registration number: 1305-546-487) and was conducted in accordance with the Helsinki Declaration. Informed consent was obtained from the patients before primary debulking surgery for use in research.

### 2.3. Cytotoxicity Assay

To determine cell viability, the 3-(4,5-dimethylthiazol-2-yl)-2,5-diphenyltetrazolium bromide (MTT) assay was employed. Both ovarian cancer cell lines (2.0 × 10^4^ cells/cm^2^), treated with 40 μM cisplatin (CDDP; EMD Millipore, Burlington, MA, USA) either alone or in combination with 40 μM ML-SI1 (Sigma Aldrich, St. Louis, MO, USA) for 48 h, were incubated with 2 mg/mL MTT solution at 37 °C for 3 h in the dark. Subsequently, dimethyl sulfoxide (DMSO) was added to dissolve the formazan crystals produced by viable cells. The absorbance was measured at 540 nm by a microplate reader (SpectraMax^®^ ABS, Molecular Devices, San Jose, CA, USA).

### 2.4. Quantitative Real-Time Polymerase Chain Reaction (qRT-PCR)

Total RNA was extracted from TOV21G and OVCAR8 cells using TRIzol reagent (Ambion, Life Technologies, Carlsbad, CA, USA) according to the manufacturer’s instructions. Reverse transcription was conducted with 500 ng of total RNA using a High-Capacity cDNA RT kit (Applied Biosystems, Thermo Fisher Scientific) on a Bio-Rad T100 thermal cycler (Bio-Rad, Hercules, CA, USA). SYBR green-based qRT-PCR was performed on a QuantStudio^TM^ 3 Real-Time PCR System (Applied Biosystems, Waltham, MA, USA) to analyze the expression of *MCOLN1* (NM_020533.3, F: 5′-TCTTCCAGCACGGAGACAAC-3′, R: 5′-GCCACATGAACCCCACAAAC-3′), *TFEB* (NM_001167827.3, F: 5′-CCAGAAGCGAGAGCTCACAGAT-3′, R: 5′-TGTGATTGTCTTTCTTCTGCCG-3′), *CTSA* (NM_000308.4, F: 5′-CAGGCTTTGGTCTTCTCTCCA-3′, R: 5′-TCACGCATTCCAGGTCTTTG-3′), *CTSD* (NM_001909.5, F: 5′-ACTGCTGGACATCGCTTGCT-3′, R: 5′-CATTCTTCACGTAGGTGCTGGA-3′). The relative abundance of mRNA was normalized to the reference gene *GAPDH* (NM_001256799.3, F: 5′-GAAGGTGAAGGTCGGAGTC-3′, R: 5′-GAAGATGGTGATGGGATTTC-3′). All primers were synthesized at Macrogen (Seoul, Republic of Korea). 

### 2.5. Western Blotting

Ovarian cancer cells were harvested and lysed with RIPA buffer, consisting of 20 mM Tris-HCl pH 8.0, 150 mM NaCl, 1 mM EDTA, 1% Triton X-100, 0.1% sodium deoxycholate, 1 mM phenylmethylsulfonyl fluoride (PMSF), 1 mM sodium orthovanadate (Na_3_VO_4_), and 1X protease inhibitor cocktail. The lysates were centrifuged at 12,000× *g*, 4 °C for 15 min, and the supernatant (whole protein) was transferred to new tubes. Protein quantification was performed using a Pierce BCA protein assay kit (Thermo Fisher Scientific). A total 10 µg of protein per well was separated using 6–10% SDS-PAGE and then transferred to 0.45 µm poly-vinylidene fluoride (PVDF) membranes (Bio-Rad). After blocking with 5% skim milk solution, the membrane was probed with primary antibodies, including TRPML1 (Sigma-Aldrich), LAMP1 (Santa Cruz Biotechnology, Dallas, TX, USA), MDR1 and Actin (Cell Signaling Technology, Danvers, MA, USA). Following incubation with secondary antibodies conjugated with horseradish peroxidase (HRP), signals were visualized using a chemiluminescence detection kit (Westar ƞC ultra 2.0, Cyanagen, Bologna, Italy). Images were captured by a chemiluminescence imaging system (Vilber, The FUSION Solo X, Eberhardzell, Germany). 

### 2.6. Sample Preparation for UHPLC/Q-TOF-MS 

Untargeted metabolites from cisplatin-resistant ovarian cancer cells (OVCAR8; 5.0 × 10^6^ cells) were extracted using an ice-cold solution consisting of 40% (*v*/*v*) acetonitrile, 40% (*v*/*v*) methanol, and 20% (*v*/*v*) H_2_O (extraction solvent). The collected ovarian cancer cells were washed with ice-cold PBS three times, and snap-frozen with liquid nitrogen. The snap-frozen cells were lysed using the extraction solvent described above and transferred to new microcentrifuge tubes. Ultrasonication in an ultrasonic bath was performed for 30 s with a subsequent rest period on ice for 30 s, totaling three cycles over 3 min. The sonicated samples were then incubated on ice for 10 min, and centrifuged at 12,000× *g* for 10 min at 4 °C. After centrifugation, the liquid phase of each sample was filtered using a 0.22 μm microfiltration membrane, and the flow-through was transferred to a screw-cap glass tube with an insert (Agilent Technologies, Santa Clara, CA, USA) for UHPLC/Q-TOF-MS analysis. 

### 2.7. Liquid Chromatography and Mass Spectrometry

Untargeted metabolite analysis was carried out on a liquid chromatograph quadrupole time-of-flight mass spectrometer (LC/Q-TOF-MS; Agilent Technologies, USA; Metabolomics Research Center for Functional Materials, Kyungsung University). For chromatographic separation (UHPLC Agilent 1290 Infinity LC system; Agilent Technologies), each sample in a volume of 1 μL was injected into ZORBAX RRHD Eclipse XDB-C18 column (2.1 × 50 mm, 1.8 μm; set temperature 30 °C; Agilent Technologies). The mobile phase A and B were 45% water with 0.1% formic acid and 55% acetonitrile with 0.1% formic acid, respectively. Gradient elution with a flow rate of 0.5 mL/min was conducted as follows: 0 min, 2% B; 1 min, 2% B; 8 min, 100% B; 10 min, 100% B; 11 min, 2% B; 20 min, 2% B. For mass spectrometry, Agilent 6545 Q-TOF/MS (Agilent Technologies) equipped with positive and negative electrospray ionization (ESI) sources was set as follows: capillary voltage 4000 V, fragmentor voltage 125 V, gas temperature 300 °C, drying gas 10 L/min, maximum pressure of nebulizer with 45 psi, sheath gas temperature 300 °C, sheath gas flow 11 L/min, and RF voltage 750 V. Data were acquired using MassHunter Software version 14.0: acquisition module version 11.0 and qualitative analysis module version 10.0 (Agilent Technologies) in both positive and negative ion modes for a full scan with a mass range from 100 to 1000 m/z. For tandem mass spectrometric detection, Agilent 6470 triple quadrupole MS/MS system was used (Agilent Technologies, USA; Metabolomics Research Center for Functional Materials, Kyungsung University).

### 2.8. Data Processing and Analysis

The Raw data files (‘-.d’) obtained from LC/Q-TOF-MS were converted to ‘-.cef’ format using Profinder 10.0 (Agilent Technologies). The converted data were further processed for peak finding, alignment, and identification in MassHunter Mass Profiler Profession 15.0. Enrichment and pathway analysis of the differentially identified metabolites was performed using MetaboAnalyst 5.0 (http://www.metaboanalyst.ca; accessed on 27 September 2023). 

### 2.9. Statistical Analysis

Data were analyzed using GraphPad Prism 9 and presented as mean ± standard error of mean (SEM) of at least three independent experiments. Normal distribution was evaluated using the Shapiro–Wilk test. For normally distributed data, we performed the unpaired *t*-test to compare two groups and one-way analysis of variance (ANOVA) to compare three or more than three categorical groups. In the case of a significant difference following one-way ANOVA, Tukey’s test was used for post hoc analysis. 

## 3. Results

### 3.1. Enhanced Lysosomal Exocytosis in Cisplatin-Resistant Ovarian Cancer Cells

To assess the resistance of ovarian cancer cells to the platinum-based drug (cisplatin), we obtained information from two ovarian cancer cell lines: TOV21G, characterized by wild type of p53 (p53-WT) and OVCAR8, characterized by mutant p53 (p53-mut) (Figure 1A). The cells were subjected to serial concentrations of cisplatin to determine their sensitivity for 48 h. As shown in Figure 1B, the IC_50_ of cisplatin was less than 10 μM in TOV21G and could not be determined in OVCAR8 (Figure 1B). Since OVCAR8 exhibited a trend of decreased cell viability at 40 μM of cisplatin, we opted to treat OVCAR8 cells with 40 μM of cisplatin in subsequent experiments. Lysosomal sequestration of chemotherapeutics with weak base is recognized as one of mechanisms contributing to chemoresistance [30]. In cisplatin-resistant cancer cells, specifically OVCAR8, there was a notable upregulation of genes associated with lysosomal biogenesis and function compared to cisplatin-sensitive cancer cells, TOV21G (Figure 1C). Mucolipin transient receptor potential channel 1 (TRPML1), encoded by *MCOLN1*, demonstrated significantly increased expression in OVCAR8 when compared to both normal cells (HK2) and cisplatin-sensitive cancer cells (TOV21G; Figure 1D). 

To evaluate the potential role of TRMPL1 in chemoresistance, we treated both TOV21G and OVCAR8 with ML-SI1 (selective inhibitor of TRMPL1) in conjunction with cisplatin. The concentration of ML-SI1 was determined following the cytotoxicity test, and subsequent experiments employed 40 μM of ML-SI1 (Appendix A). Pharmacological inhibition of TRMPL1 sensitized OVCAR8 cells to cisplatin exposure, leading to increased cell death (Figure 1E). Additionally, we accomplished the deletion of *MCOLN1* using the dual sgRNA CRISPR-Cas9 system. The efficacy of TRPML1 knockdown was validated through Western blotting (Figure 1F). Consistent with the outcomes of pharmacological inhibition, genetic inhibition of *MCOLN1* also enhanced cisplatin-mediated cell death (Figure 1G). LAMP1 is a lysosomal membrane protein that contributes to the fusion of lysosomes with the plasma membrane for lysosomal exocytosis [31]. In addition, TRPML1, a calcium channel present in the lysosomal membrane, induces lysosomal exocytosis through calcium regulation [32,33]. To investigate whether TRPML1-mediated lysosomal exocytosis is involved in the exclusion of cisplatin outside of the cells, we examined the localization of LAMP1 expression after treating both cell lines with cisplatin and ML-SI1. Exposure to ML-SI1 markedly reduced total LAMP1 expression in OVCAR8, but not in TOV21G (Figure 1H). To observe spatial changes in LAMP1 induced by cisplatin or ML-SI1, we measured the localization of LAMP1 using confocal microscopy. As we expected, TOV21G displayed a scattered pattern of LAMP1 by cisplatin treatment within the intracellular compartment, whereas OVCAR8 exhibited an arrangement of LAMP1 along the plasma membrane by cisplatin treatment (Figure 1I). On the other hand, the inhibition of TRPML1 using ML-SI1 disrupted the localization of LAMP1 along the plasma membrane induced by cisplatin treatment (Figure 1I), which means TRPML1 inhibition interrupted drug efflux conducted by lysosomal exocytosis. Based on these findings, it is plausible that the reduced sensitivity of cisplatin-resistant ovarian cancer cells to cisplatin exposure results from TRPML1-mediated lysosomal exocytosis. 

### 3.2. Untargeted Metabolomic Profile Identified by Treatment with Cisplatin and ML-SI1 in the Cisplatin-Resistant Ovarian Cancer Cells

The metabolomic impact of TRPML1 inhibition on cisplatin-resistant ovarian cancer cells was assessed using a UPLC/Q-TOF-MS. Untargeted metabolomic analysis was conducted on four groups, comprising vehicle-, CDDP alone-, ML-SI1 alone-, and CDDP with ML-SI1-treated samples (Figure 2A). A total of 1446 metabolites were identified as differentially altered by treating cisplatin with or without TRPML1 inhibitor. Principal component analysis (PCA) score plots for ESI-positive and ESI-negative modes revealed a distinct separation between ML-SI1-treated samples and untreated samples (Figure 2B). Hierarchical condition trees demonstrated that the four sample groups were clustered, particularly with the treatment of ML-SI1 in both ESI-positive and ESI-negative modes (Figure 2C). 

Among the annotated metabolites, our attention was focused on the compounds in the superclasses of organic acids, nucleic acids, lipids, and fatty acyls, which were frequently found (Table 1). The heatmap also illustrated that the four groups were categorized with the treatment of ML-SI1 (Figure 3A). To identify biologically meaningful patterns enriched in quantitative metabolomic data, metabolite set enrichment analysis was performed. The results of the enrichment analysis for metabolites revealed the top 25 sets, including arginine and proline metabolism, arginine biosynthesis, another amino acid metabolism, fructose, and mannose metabolism, and TCA cycle (Figure 3B). The lipid enrichment analysis displayed the top 23 sets, including monoacylglycerophosphoiositols, monoalkylamines, and amino fatty acids (Figure 3C). To elucidate functional roles, metabolites were mapped to KEGG pathways. The results of human KEGG pathways were plotted to delineate the most significant metabolic pathways. The top eight pathways that emerged with low *p*-values are indicated in Figure 3D: (1) Alanine, aspartate, and glutamate metabolism, (2) Arginine and proline metabolism, (3) Aminoacyl-tRNA biosynthesis, (4) Purine metabolism, (5) Arginine biosynthesis, (6) Histidine metabolism, (7) Taurine and hypotaurine metabolism, and (8) Butanoate metabolism. Overall, TRPML1 inhibition induced distinct metabolite changes associated with various metabolic pathways in the cisplatin-resistant ovarian cancer cells. 

### 3.3. The Alteration of Arginine by Pharmacological Inhibition of TRPML1

Based on untargeted metabolomic data, four metabolites, including arginine, glutamic acid, cysteine, and creatine, were selected. To validate the untargeted metabolomic data, targeted metabolomic analysis was carried out using an Agilent 6470 triple quadrupole LC-MS/MS system. Pharmacological inhibition of TRPML1 using ML-SI1 resulted in decreased intracellular contents of arginine, glutamic acid, cysteine, and creatine (Figure 4A). Additionally, we examined the alteration of these metabolites by TRPML1 inhibition in ovarian cancer patient-derived cells (A8 and A39). A8 and A39 cells had been isolated and established from the ascites of two ovarian cancer patients, respectively. The established ovarian cancer cells, A8 and A39, were also found to be resistant to cisplatin (Figure 4B). Triple quadrupole LC-MS/MS analysis in A8 and A39 revealed decreased intracellular contents of the measured metabolites, including arginine, glutamic acid, cysteine, and creatine, in response to cisplatin and ML-SI1 (Figure 4C,D). To validate our targeted metabolomic analysis, we subjected OVCAR8 cells to exogenous arginine treatment and indirectly evaluated lysosomal exocytosis by confirming the expressions of LAMP1 and TRPML1. The exogenous supplementation of arginine led to an overall increase in LAMP1 expression, and the pharmacological inhibition of TRPML1 alleviated the arginine-induced elevation of total LAMP1 expression. Interestingly, the expression of TRPML1 remained unchanged in response to exogenous arginine (Figure 4E). Taken together, these findings imply that arginine potentially contributes to TRPML1-mediated chemoresistance in ovarian cancer cells. 

## 4. Discussion

Our findings demonstrated that lysosomal exocytosis contributed to resistance to cisplatin in ovarian cancer cells. Inhibition of lysosomal calcium channel attenuated lysosomal exocytosis and sensitized drug-resistant ovarian cancer cells (OVCAR8) to cisplatin treatment. Metabolomic analysis provided a specific profile of cisplatin-resistant cancer cells after sensitization to cisplatin through the inhibition of lysosomal exocytosis. Altered metabolites across superclasses, including organic acids, nucleic acids, lipids, and fatty acyls, revealed KEGG metabolic and lipid pathways contributing to drug sensitization. Targeted metabolomic analysis showed that concentrations of arginine, glutamic acid, cysteine, and creatine decreased in response to the inhibitor of lysosomal calcium channel TRPML1 in the cisplatin-resistant cancer cell line and two types of patient-derived ovarian cancer cells. 

Recently, the role of cancer lysosomes has gained attention as a compelling strategy to sensitize cancer cells to chemotherapy. Enhanced lysosomal exocytosis confers robust invasiveness and chemoresistance to human cancer cells [34]. The process of lysosomal exocytosis is calcium-dependent, and lysosomal secretion occurs in response to an increase in intracellular free calcium concentration, facilitated by the fusion of the lysosomal membrane with the plasma membrane [35]. Given the dependency of lysosomal exocytosis on calcium, it can be suggested that TRPML1, a nonselective calcium-permeable cation channel present in the lysosomal membrane, is implicated in the process of lysosomal exocytosis through Ca^2+^ regulation [32,33]. According to our findings, the cisplatin-resistant ovarian cancer cell line, OVCAR8, exhibited higher expression levels of TRPML1 compared to both the cisplatin-sensitive cancer cell line (TOV21G) and non-cancerous cells. These observations led us to focus on TRPML1 as a potential target for overcoming chemoresistance. Interestingly, a specific inhibitor of TRPML1 hindered the efflux of cisplatin by blocking the fusion of lysosomal and plasma membranes, subsequently promoting cell death in cisplatin-resistant OVCAR8 cells. Accumulating evidence supports TRPML1 as an interesting target for cancer. Bladder urothelial carcinoma and head and neck squamous cell carcinoma exhibit a high correlation between TRPML1 expression and oncogenic HRAS (mutant HRAS). Selective inhibition of TRPML1 or Knockdown of *MCOLN1* reduces cancer proliferation by disrupting oncogenic HRAS clustering localized in the plasma membrane [19]. Additionally, deficiency of *MCOLN1* and TRPML1 hinders the growth of patient-derived melanoma cells through the interruption of macropinocytosis and depletion of serine in both in vitro and in vivo experiments [21]. It has been reported that TRPML1 is partially involved in resistance to chemotherapy agents. The overexpression of transmembrane member 16A (TMEM16A) leads to cisplatin resistance, and TRPML1 knockdown diminishes TMEM16A overexpression, resulting in sensitization to cisplatin treatment in head and neck squamous cell carcinoma cells [36,37]. In ovarian cancer, abundant lysosomes and activated cathepsin D found in cisplatin-resistant SKOV3/DDP cells are required for the maintenance of autophagic flux, partially involved in the promotion of cisplatin resistance [38]. However, there has been little evidence supporting the contribution of TRPML1 to chemoresistance in ovarian cancer. In this study, we first demonstrated that TRPML1 inhibition increased vulnerability to chemoresistance in ovarian cancer cells to cisplatin treatment. Furthermore, our metabolomic analysis provided evidence suggesting a potential association between arginine and TRPML1-mediated chemoresistance. In patient-derived ovarian cancer cells exhibiting cisplatin resistance, the inhibition of TRPML1 was observed to reduce the cellular content of arginine, which had been elevated by cisplatin treatment, as determined through MS/MS analysis. Interestingly, TRPML1 inhibition effectively mitigated the upregulation of LAMP1 expression induced by the addition of exogenous arginine, while arginine itself did not exert any discernible influence on TRPML1 expression. These findings collectively propose that TRPML1 signaling may precede arginine-induced lysosomal exocytosis. 

Arginine, a semi-essential amino acid, plays a crucial role in the growth and migration of cancer cells, particularly characterized by the development of chemoresistance and unfavorable clinical outcomes [39]. Several cancers, including ovarian cancer, exhibit arginine auxotrophy, a feature marked by defective arginine synthesis due to silencing of argininosuccinate synthetase 1 (ASS1) or arininosuccinate lyase (ASL). Consequently, these cancers rely on the arginine supply from the extracellular compartment [40,41,42]. Thus, arginine deprivation could be a therapeutic target for arginine-auxotrophic cancers. In the human breast cancer cell line MDA-MB-231, arginine starvation induces cell death by depleting aspartate and disrupting the malate-aspartate shuttle [43]. Ovarian cancer cell lines with arginine auxotrophy are vulnerable to the treatment of recombinant human arginase I cobalt, leading to caspase-independent and non-apoptotic cell death [40]. In lysosomes, arginine is transported to the lysosomal lumen by solute carrier family 7A1 (SLC7A1). The amplified expression of lysosomal arginine transporter SLC7A1 is frequently found in various solid tumors, including hepatocarcinoma, colorectal cancer, breast cancer, and ovarian cancer [44,45,46,47,48]. In ovarian cancer, an elevated expression of the arginine transporter SLC7A1 in tumor tissue is known to be correlated with a poor survival outcome for patients [48]. Moreover, arginine depletion weakens mTORC1-mediated autophagic signaling, consequently showing a decrease in cell division and migration of ovarian cancer cell line [49]. In this study, untargeted metabolomic analysis suggests that TRPML1 inhibition in cisplatin-resistant ovarian cancer cells induces alterations in metabolites associated with arginine and other amino acid metabolisms. Our targeted metabolomic analysis indicates the intracellular increase of arginine induced by cisplatin treatment is exhausted by pharmacological inhibition of TRPML1 in ovarian cancer cell line and patient-derived cancer cells. These findings collectively offer compelling evidence for the connection between TRPML1 and arginine depletion in cancer cells, particularly in the context of chemoresistance. However, it is important to note that our current understanding is limited in determining whether TRPML1 acts on arginine auxotrophy or the lysosomal arginine transporter. Therefore, further studies are warranted to elucidate the more specific molecular mechanisms that underlie the link between TRPML1 inhibition and arginine deprivation, potentially providing valuable insights for the development of targeted therapies in overcoming chemoresistance in ovarian cancer. 

## 5. Conclusions

This study underscores the significance of TRPML1-mediated lysosomal exocytosis in modulating chemoresistance in ovarian cancer. The lysosome’s central role in drug sequestration has positioned it as a promising target for overcoming resistance to chemotherapy. By targeting TRPML1 through pharmacological and genetic interventions, this research successfully disrupted lysosomal exocytosis in cisplatin-resistant ovarian cancer cells. This interruption sensitized the ovarian cancer cells to cisplatin treatment, offering a potential breakthrough in overcoming resistance mechanisms. Based on our metabolomic analysis, the introduction of exogenous arginine amplified lysosomal exocytosis. Collectively, these findings deepen our understanding of the intricate interplay between TRPML1-mediated lysosomal exocytosis and chemoresistance. The identified metabolites and pathways provide a foundation for innovative therapeutic strategies, potentially revolutionizing the approach to combating chemoresistance in ovarian cancer. This research contributes valuable insights that could shape the development of targeted interventions, offering hope for improved treatment outcomes in ovarian cancer patients.

## Figures and Tables

**Figure 1 cells-13-00115-f001:**
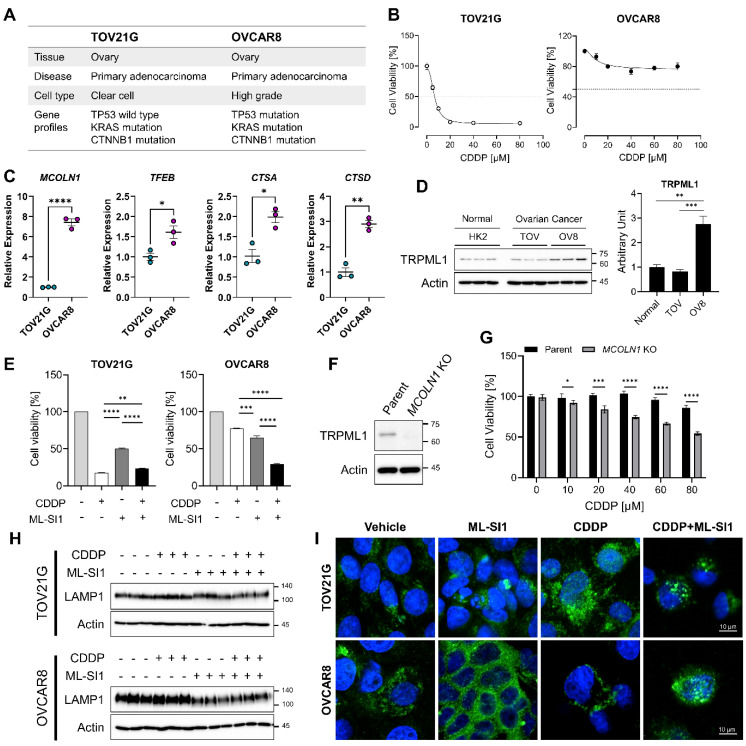
Sensitization of cisplatin-resistant ovarian cancer cells by TRPML1. (**A**) Characterization of TOV21G and OVCAR8. (**B**) Assessment of cell viability in cisplatin-sensitive (TOV21G) and -resistant cancer cells (OVCAR8) following serial concentrations of cisplatin (CDDP) treatment, determined by MTT assay. (**C**) Evaluation of gene expression associated with lysosomal biogenesis and regulation by qRT−PCR. (**D**) Comparison of basal levels of TRPML1 expression in normal cells, cisplatin-sensitive, and -resistant ovarian cancer cells using Western blotting. (**E**) Determination of sensitization of ovarian cancer cells to cisplatin (10 µM in TOV21G, 40 µM in OVCAR8) by 40 µM of ML-SI1. (**F**) Confirmation of *MCOLN1* knockdown by Western blotting. (**G**) Evaluation of sensitization of cells to cisplatin by genetic inhibition of *MCOLN1*. (**H**) Assessment of LAMP1 expression by Western blotting after treatment with cisplatin with or without ML-SI1. (**I**) Detection of exocytosis by spatial changes in LAMP1 (Green) using a confocal microscope. The blue signal represents DAPI (nucleus). Data are presented as mean ± SEM (* *p* < 0.05, ** *p* < 0.01, *** *p* < 0.001, **** *p* < 0.0001).

**Figure 2 cells-13-00115-f002:**
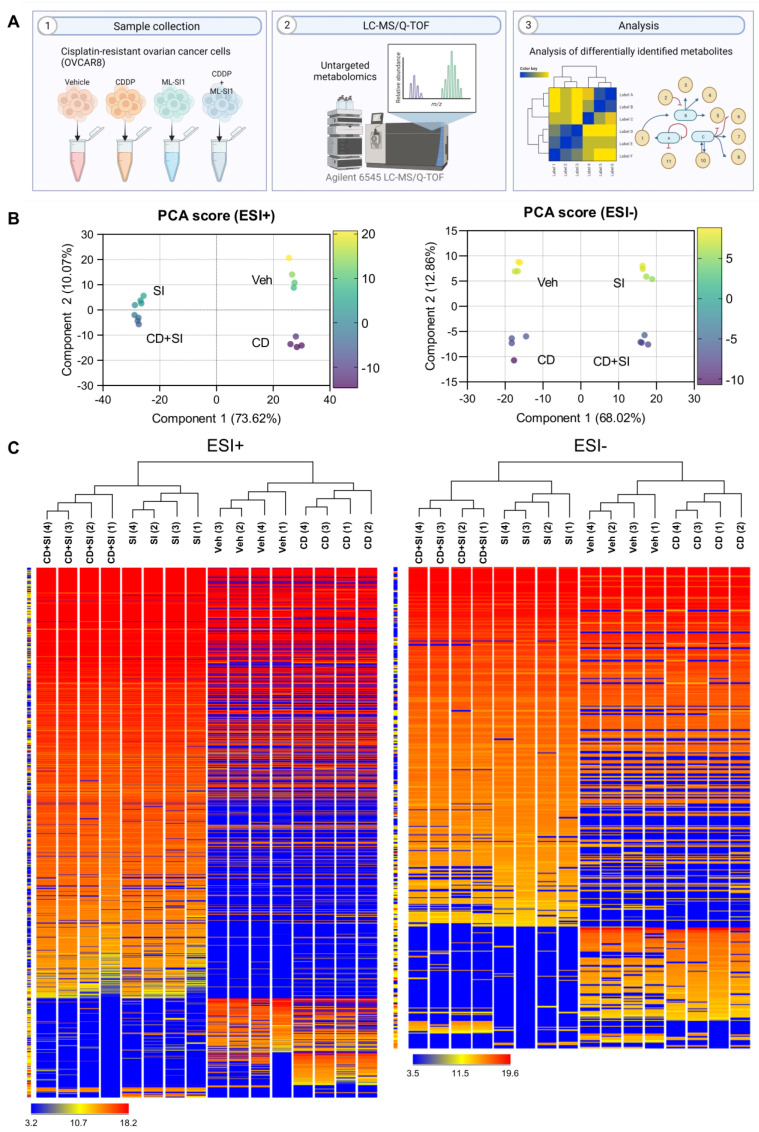
Untargeted metabolites from cisplatin-resistant ovarian cancer cells in response to vehicle, cisplatin (40 µM; CD), ML-SI1 (40 µM; SI), and a combination of cisplatin and ML-SI1. (**A**) A schematic diagram illustrating the untargeted metabolomics approach using LC−MS/Q−TOF. (**B**) Principal component analysis (PCA) score plots for ESI-positive and ESI-negative modes. (**C**) Hierarchical condition trees representing the relationship among the four sample groups in both ESI-positive and ESI-negative modes.

**Figure 3 cells-13-00115-f003:**
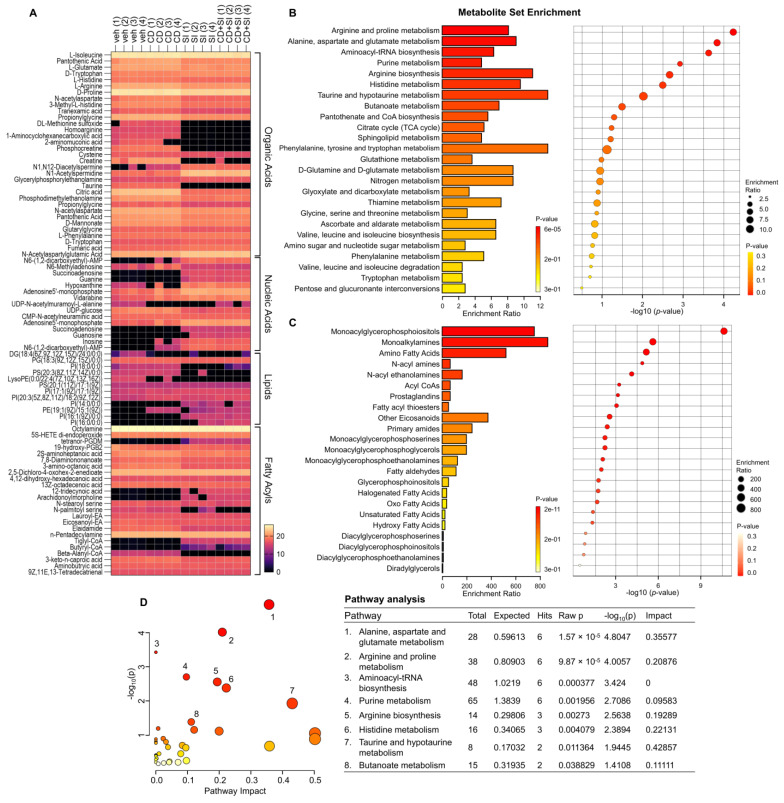
Metabolomic analysis revealing alterations induced by TRPML1 inhibition in cisplatin−resistant ovarian cancer cells. (**A**) Heatmap depicting variations in organic acids, nucleic acids, lipids, and fatty acyls. (**B**) Metabolite set enrichment analysis highlighting the top 25 sets. (**C**) Lipid set enrichment analysis presenting the top 23 sets. (**D**) Human KEGG pathways of the differentially identified metabolites. Color indicates the levels of significance (−log_10_(*p*)) from yellow to red. The size of circle indicates the pathway impact.

**Figure 4 cells-13-00115-f004:**
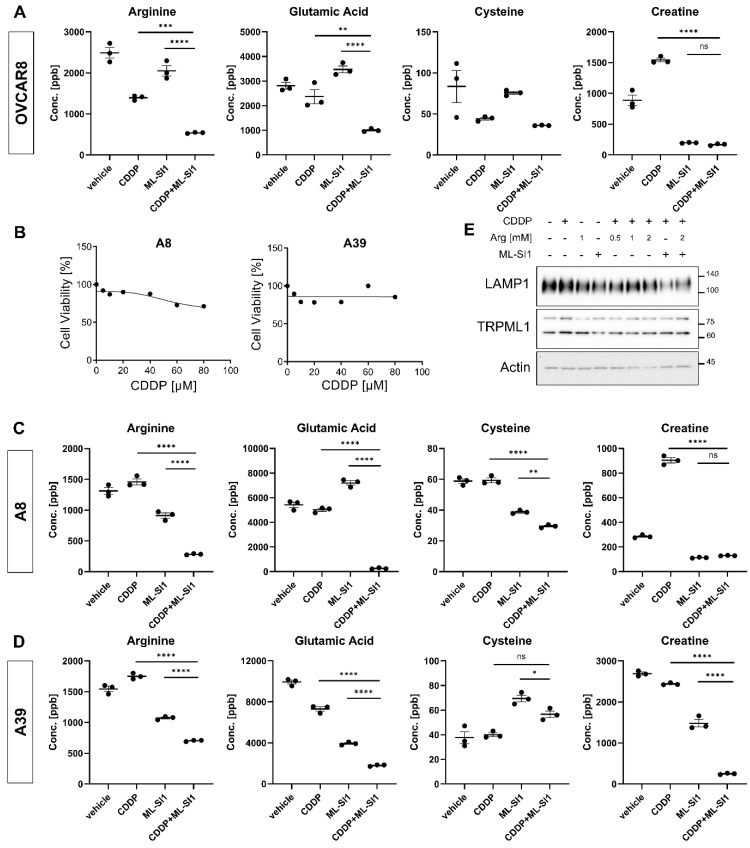
Downregulation of arginine contents by the combination of cisplatin with TRPML1 inhibition. (**A**) Targeted metabolomic analysis of arginine, glutamic acid, cysteine, and creatine in OVCAR8 cells using an Agilent 6470 triple quadrupole LC-MS/MS. (**B**) Assessment of cisplatin−resistance in ovarian cancer patient-derived cells, A8 and A39. (**C**,**D**) Targeted metabolomic analysis of arginine, glutamic acid, cysteine, and creatine in A8 and A39 cells using an Agilent 6470 triple quadrupole LC-MS/MS. (**E**) Determination of alterations in LAMP1 and TRPML1 expression by exogenous addition of arginine. Data are presented as mean ± SEM (* *p* < 0.05, ** *p* < 0.01, *** *p* < 0.001, **** *p* < 0.0001).

**Table 1 cells-13-00115-t001:** Metabolites differentially identified in the treatment groups of vehicle (Veh), cisplatin (CD) alone, ML-SI1 (SI1) alone, and combination of cisplatin and ML-SI1.

Super Class	Compound	*p*	*p* (corr)	Log FC					
				CD Vs. Veh		SI1 vs. Veh		CD + SI1 vs. Veh	
Organic Acids	L-Isoleucine	6.1 × 10^−4^	7.2 × 10^−3^	0.098259	up	−0.263855	down	−0.372273	down
	Pantothenic Acid	2.4 × 10^−5^	4.2 × 10^−4^	0.661634	up	−1.050560	down	−0.864870	down
	L-Glutamate	1.9 × 10^−4^	2.6 × 10^−3^	0.040453	up	−0.937819	down	−0.975943	down
	D-Tryptophan	1.2 × 10^−5^	2.2 × 10^−4^	0.059481	up	0.646954	up	0.570774	up
	L-Histidine	1.5 × 10^−7^	3.3 × 10^−6^	−0.010078	down	−0.877106	down	−0.872246	down
	L-Arginine	2.7 × 10^−9^	6.7 × 10^−8^	0.063723	up	−1.192158	down	−1.062908	down
	D-Proline	3.8 × 10^−7^	8.1 × 10^−6^	−0.161091	down	−1.120415	down	−1.136900	down
	N-acetylaspartate	8.8 × 10^−6^	1.6 × 10^−4^	0.593977	up	−1.343704	down	−1.137489	down
	3-Methyl-L-histidine	3.0 × 10^−9^	7.3 × 10^−8^	0.169186	up	−1.319733	down	−1.313189	down
	Tranexamic acid	3.0 × 10^−6^	5.8 × 10^−5^	0.339138	up	−1.268448	down	−1.396981	down
	Propionylglycine	2.7 × 10^−11^	7.2 × 10^−10^	0.088356	up	−1.694998	down	−1.763273	down
	DL-Methionine sulfoxide	1.4 × 10^−4^	2.1 × 10^−3^	3.855974	up	−12.244130	down	−12.295345	down
	Homoarginine	6.7 × 10^−21^	7.6 × 10^−18^	0.213843	up	−15.487052	down	−15.538267	down
	1-Aminocyclohexanecarboxylic acid	8.9 × 10^−19^	1.3 × 10^−16^	0.337267	up	−15.848567	down	−15.899782	down
	2-aminomuconic acid	9.8 × 10^−4^	1.0 × 10^−2^	−7.809795	down	−16.596827	down	−16.648043	down
	Phosphocreatine	7.7 × 10^−20^	4.1 × 10^−17^	0.332651	up	−18.390627	down	−18.441843	down
	Cysteine	3.4 × 10^−6^	6.6 × 10^−5^	−0.391922	down	−0.808666	down	−1.392947	down
	Creatine	1.3 × 10^−5^	2.3 × 10^−4^	1.138592	up	−19.575426	down	−15.563755	down
	N1,N12-Diacetylspermine	2.9 × 10^−4^	3.7 × 10^−3^	10.575350	up	14.352155	up	13.833116	up
	N1-Acetylspermidine	2.2 × 10^−10^	5.8 × 10^−9^	0.014311	up	4.507406	up	4.032513	up
	Glycerylphosphorylethanolamine	6.6 × 10^−8^	1.4 × 10^−6^	0.089578	up	1.130651	up	0.971383	up
	Taurine	1.5 × 10^−21^	2.8 × 10^−18^	0.060276	up	−17.762808	down	−17.814024	down
	Phosphocreatine	1.3 × 10^−4^	2.2 × 10^−3^	-4.279326	down	−17.412300	down	−17.421219	down
	Citric acid	4.4 × 10^−15^	2.0 × 10^−13^	0.243635	up	−4.654179	down	−4.469467	down
	Phosphodimethylethanolamine	9.1 × 10^−8^	2.7 × 10^−6^	0.259087	up	−1.917299	down	−1.369122	down
	Propionylglycine	1.9 × 10^−7^	5.2 × 10^−6^	0.288090	up	−1.634593	down	−1.292515	down
	N-acetylaspartate	9.4 × 10^−11^	3.6 × 10^−9^	0.059454	up	−1.350250	down	−1.221140	down
	Pantothenic Acid	8.5 × 10^−13^	3.6 × 10^−11^	0.322907	up	−1.223236	down	−1.124733	down
	D-Mannonate	1.1 × 10^−9^	4.1 × 10^−8^	0.075493	up	−0.950808	down	−0.849545	down
	Glutarylglycine	6.1 × 10^−10^	2.2 × 10^−8^	0.195900	up	−0.812933	down	−0.372046	down
	L-Phenylalanine	5.7 × 10^−6^	1.3 × 10^−4^	0.174341	up	0.503141	up	0.551741	up
	D-Tryptophan	1.3 × 10^−9^	5.0 × 10^−8^	0.010935	up	0.824541	up	0.780550	up
	Fumaric acid	4.2 × 10^−9^	1.4 × 10^−7^	0.237362	up	0.990431	up	0.987371	up
	N-Acetylaspartylglutamic Acid	2.9 × 10^−11^	1.1 × 10^−9^	0.096752	up	1.587130	up	1.543385	up
Nucleic Acids	N6-(1,2-dicarboxyethyl)-AMP	9.3 × 10^−5^	1.4 × 10^−3^	6.615269	up	17.624231	up	17.374481	up
	N6-Methyladenosine	3.5 × 10^−3^	3.3 × 10^−2^	2.346935	up	−0.535506	down	−0.893127	down
	Inosine	1.3 × 10^−4^	1.9 × 10^−3^	10.738907	up	16.922800	up	16.609482	up
	Succinoadenosine	7.6 × 10^−18^	4.0 × 10^−16^	−0.296469	down	15.327309	up	15.280434	up
	Guanine	2.7 × 10^−14^	7.4 × 10^−13^	−0.296469	down	14.768631	up	15.234587	up
	Hypoxanthine	2.3 × 10^−3^	2.3 × 10^−2^	−9.678612	down	4.731751	up	4.096037	up
	Adenosine5′-monophosphate	1.6 × 10^−5^	2.8 × 10^−4^	1.216438	up	3.822020	up	3.535231	up
	Vidarabine	2.9 × 10^−4^	3.7 × 10^−3^	1.676630	up	2.831606	up	2.738785	up
	UDP-N-acetylmuramoyl-L-alanine	1.2 × 10^−4^	2.0 × 10^−3^	−13.744226	down	−13.832284	down	−3.610789	down
	UDP-glucose	2.5 × 10^−7^	7.0 × 10^−6^	0.263239	up	−2.822538	down	−2.443968	down
	CMP-N-acetylneuraminic acid	2.5 × 10^−6^	5.9 × 10^−5^	0.100336	up	−0.336864	down	−0.310667	down
	Adenosine5’-monophosphate	5.8 × 10^−6^	1.3 × 10^−4^	1.007034	up	3.759728	up	3.568693	up
	Succinoadenosine	5.7 × 10^−21^	4.5 × 10^−19^	0.021034	up	14.439548	up	14.400873	up
	Guanosine	1.3 × 10^−4^	2.1 × 10^−3^	0.021034	up	11.394004	up	15.251740	up
	Inosine	2.4 × 10^−4^	3.5 × 10^−3^	7.638305	up	17.426504	up	17.101746	up
	N6-(1,2-dicarboxyethyl)-AMP	2.2 × 10^−5^	4.7 × 10^−4^	10.120125	up	17.931795	up	17.691513	up
Lipids	DG(18:4(6Z,9Z,12Z,15Z)/24:0/0:0)	1.7 × 10^−3^	1.7 × 10^−2^	−9.790112	down	−9.303253	down	−9.187818	down
	PG(18:3(9Z,12Z,15Z)/0:0)	2.9 × 10^−3^	2.8 × 10^−2^	0.21228218	up	−0.14583015	down	−0.03360176	down
	PI(18:0/0:0)	3.0 × 10^−6^	5.7 × 10^−5^	1.0861263	up	−13.140982	down	−7.942805	down
	PS(20:3(8Z,11Z,14Z)/0:0)	1.4 × 10^−4^	2.0 × 10^−3^	0.32224178	up	−10.7650585	down	−14.509828	down
	LysoPE(0:0/22:4(7Z,10Z,13Z,16Z))	2.1 × 10^−4^	3.0 × 10^−3^	−11.819713	down	−15.543326	down	−15.552244	down
	PS(20:1(11Z)/17:1(9Z))	6.8 × 10^−4^	8.8 × 10^−3^	−0.20365238	down	−0.11919689	down	−0.6867838	down
	PI(17:1(9Z)/17:1(9Z))	5.8 × 10^−4^	7.8 × 10^−3^	0.84307194	up	1.2043552	up	1.0318031	up
	PI(20:3(5Z,8Z,11Z)/18:2(9Z,12Z))	4.1 × 10^−7^	1.1 × 10^−5^	1.6445503	up	0.9564352	up	1.790739	up
	PI(14:0/0:0)	2.2 × 10^−4^	3.2 × 10^−3^	0.02103369	up	12.9155	up	10.260044	up
	PE(19:1(9Z)/15:1(9Z))	6.7 × 10^−5^	1.3 × 10^−3^	13.923148	up	8.844269	up	13.880317	up
	PI(16:1(9Z)/0:0)	8.3 × 10^−16^	4.0 × 10^−14^	0.02103369	up	14.161881	up	14.302684	up
	PI(16:0/0:0)	7.9 × 10^−4^	9.9 × 10^−3^	0.02103369	up	7.230421	up	14.790086	up
Fatty Acyls	5S-HETE di-endoperoxide	5.3 × 10^−4^	6.4 × 10^−3^	−0.12212753	down	−0.22711945	down	−0.3006935	down
	tetranor-PGDM	2.8 × 10^−7^	6.1 × 10^−6^	−0.2964691	down	11.25654	up	13.056505	up
	19-hydroxy-PGB2	2.3 × 10^−3^	2.3 × 10^−2^	−1.8916245	down	−1.4892006	down	−2.0951862	down
	2S-aminoheptanoic acid	1.2 × 10^−4^	1.8 × 10^−3^	0.5473728	up	−0.85133743	down	−0.87155724	down
	7,8-Diaminononanoate	1.3 × 10^−6^	2.7 × 10^−5^	0.3862648	up	−0.7941799	down	−0.745224	down
	3-amino-octanoic acid	1.7 × 10^−8^	4.2 × 10^−7^	0.30552673	up	−1.9246368	down	−1.9378643	down
	2,5-Dichloro-4-oxohex-2-enedioate	6.5 × 10^−4^	7.6 × 10^−3^	−0.1428318	down	0.16801453	up	0.17915154	up
	4,12-dihydroxy-hexadecanoic acid	4.4 × 10^−3^	4.0 × 10^−2^	0.27423096	up	0.010660172	up	−0.35116768	down
	13Z-octadecenoic acid	3.9 × 10^−5^	6.5 × 10^−4^	−0.32780457	down	−0.46811676	down	−0.7879505	down
	12-tridecynoic acid	1.6 × 10^−4^	2.2 × 10^−3^	−0.2964691	down	11.032591	up	15.214987	up
	Arachidonoylmorpholine	1.4 × 10^−4^	2.1 × 10^−3^	−0.2964691	down	10.467652	up	14.605372	up
	N-stearoyl serine	4.0 × 10^−3^	3.7 × 10^−2^	−0.48096085	down	−0.9363117	down	−1.4242802	down
	N-palmitoyl serine	9.6 × 10^−5^	1.4 × 10^−3^	−0.49995232	down	−9.52323	down	−15.441587	down
	Lauroyl-EA	1.6 × 10^−3^	1.7 × 10^−2^	0.07388115	up	−0.33638382	down	−0.5638695	down
	Eicosanoyl-EA	6.5 × 10^−4^	7.6 × 10^−3^	−0.52410316	down	−0.7610588	down	−1.0543919	down
	Elaidamide	2.2 × 10^−10^	5.7 × 10^−9^	0.079538345	up	−2.5760689	down	−2.5199966	down
	n-Pentadecylamine	3.1 × 10^−4^	3.9 × 10^−3^	−0.13066101	down	−0.31131744	down	−0.26801872	down
	Tiglyl-CoA	8.5 × 10^−18^	4.2 × 10^−16^	−0.2964691	down	14.160582	up	14.233025	up
	Butyryl-CoA	9.1 × 10^−5^	1.4 × 10^−3^	−0.2964691	down	8.490838	up	6.2274776	up
	Beta-Alanyl-CoA	1.2 × 10^−18^	6.4 × 10^−17^	−0.45971966	down	−14.203965	down	−14.212883	down
	3-keto-n-caproic acid	1.7 × 10^−10^	6.6 × 10^−9^	0.44731903	up	−1.2083969	down	−0.72130203	down
	Ƴ-Aminobutryic acid	3.7 × 10^−6^	8.6 × 10^−5^	0.117744446	up	−0.28038788	down	−0.47836876	down
	9Z,11E,13-Tetradecatrienal	6.6 × 10^−4^	8.6 × 10^−3^	−0.09829712	down	−0.016868591	down	−0.27725792	down

## Data Availability

The data presented in this study are available from the corresponding author upon request.

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
