# Peer review of "Modulation of Cisplatin Sensitivity through TRPML1-Mediated Lysosomal Exocytosis in Ovarian Cancer Cells: A Comprehensive Metabolomic Approach"

_cells, 2024, doi:10.3390/cells13020115_

Round 1

Reviewer 1 Report

Comments and Suggestions for Authors

Comments to the authors:
This manuscript explores the role of TRPML1-mediated lysosomal exocytosis in the chemoresistance of ovarian cancer cells. Overall, the idea is interesting and would add valuable information to the existing literature. Therefore, I would like to suggest some modifications to improve its quality:

1. The global incidence of ovarian cancer or the incidence of ovarian cancer compared to other gynaecological cancers could be noted in the introduction to emphasise the clinical significance of the research in this field.
2. Moreover, the 5-year survival rate of ovarian cancer should be added to the introduction.
3. The introduction lacks detailed information about the idea of the study; e.g., in the introduction, the authors should note why they investigated the lysosomal calcium channel (TRPML) 1 and not the other members of this family. What is the specific role of TRPML1?
4. Also, different ion channels have been identified in the lysosome. The authors should explain why they focused on the calcium channels. What are the functions of calcium channels, and what is the role of lysosomal calcium in cellular functions?
5. In lines 75–76, the authors should be detailed on the metabolomic profiles of changes in platinum-sensitive and platinum-resistant ovarian cancer cells.
6. In line 92, the culture duration should be noted.
7. In line 95, the number of patients should be mentioned.
8. In line 102, the concentration of cells for the MTT assay should be mentioned in the methodology. The same comments for all tests.
9. In lines 104–106, the authors should mention the dose of cisplatin and ML-SI1. Also, how did they reach the selected concentration of cisplatin, ML-SI1, and 48-hour exposure time? whether they did dose response or time response tests or chose the optimum dose based on the previous studies. In addition, the concentration of cells in treated and non-treated groups should be noted here. While some of this information is provided in the results, the method is vague.

Author Response

We would like to express our gratitude to the reviewers for taking time to assess the acceptability of our manuscript and for providing helpful suggestions. We carefully considered their concerns while revising the manuscript, and we believe that the revised version is an improved one. All changes have been highlighted in red in the revised manuscript. Our responses to the specific comments from the reviewers are provided below.

Comment 1. The global incidence of ovarian cancer or the incidence of ovarian cancer compared to other gynaecological cancers could be noted in the introduction to emphasise the clinical significance of the research in this field.

Response 1: Thank you for the comment. We have added the contents that reviewer pointed out in the introduction (lines 41, 55 - 59).

Comment 2. Moreover, the 5-year survival rate of ovarian cancer should be added to the introduction.

Response 2: Thank you for the comment. We have added the contents that reviewer pointed out in the introduction (lines 42 – 44, lines 55 – 59).

Comment 3. The introduction lacks detailed information about the idea of the study; e.g., in the introduction, the authors should note why they investigated the lysosomal calcium channel (TRPML) 1 and not the other members of this family. What is the specific role of TRPML1?

Response 3: TRPML1 plays a crucial role in the regulation of lysosomal trafficking, including exocytosis, achieved through the release of calcium from lumen into the cytosol. Our study suggests a significant involvement of lysosomal exocytosis in cisplatin sequestration, potentially contributing to the chemoresistance observed in ovarian cancer cells. Considering this, we believe that manipulating lysosomal exocytosis, specifically by targeting TRPML1, could present an innovative approach to overcoming chemoresistance in cancer treatment. Additional content has been integrated into the introduction for further clarification (lines 73 – 87).

Comment 4. Also, different ion channels have been identified in the lysosome. The authors should explain why they focused on the calcium channels. What are the functions of calcium channels, and what is the role of lysosomal calcium in cellular functions?

Response 4: Our primary focus is on the involvement of lysosomal exocytosis in the development of chemoresistance, particularly through drug sequestration. Lysosomal exocytosis is identified as a calcium-dependent process. Notably, previous research has demonstrated that the activation of TRPML1 induces lysosomal exocytosis, leading to an enhanced lysosomal clearance. Additionally, there is already evidence of TRPML1 overexpression in various tumor types, including head and neck carcinoma, breast cancer, melanoma, and others. As a strategic approach to overcoming chemoresistance, we selected TRPML1 as a target, aiming to reduce the efflux of cisplatin from the cells. These pertinent details are elaborated in the discussion section (lines 15-25).

Comment 5. In lines 75–76, the authors should be detailed on the metabolomic profiles of changes in platinum-sensitive and platinum-resistant ovarian cancer cells.

Response 5: As a reviewer’s suggestion, we have added more details (lines 96 – 100).

Comment 6. In line 92, the culture duration should be noted.

Response 6: The details about the culture duration were added on lines 116 - 118.

Comment 7. In line 95, the number of patients should be mentioned.

Response 7: A8 and A39 cell lines isolated from two patients were utilized in this study (line 121).

Comment 8. In line 102, the concentration of cells for the MTT assay should be mentioned in the methodology. The same comments for all tests.

Response 8: The concentration and total number of cells were presented on lines 117- 118, 130 and 176.

Comment 9. In lines 104–106, the authors should mention the dose of cisplatin and ML-SI1. Also, how did they reach the selected concentration of cisplatin, ML-SI1, and 48-hour exposure time? whether they did dose response or time response tests or chose the optimum dose based on the previous studies. In addition, the concentration of cells in treated and non-treated groups should be noted here. While some of this information is provided in the results, the method is vague.

Response 9: The concentration of cisplatin was determined by analyzing cell viability data obtained through the MTT assay (refer to Figure 1B and lines 238 - 240). Given that the cytotoxic response to cisplatin manifested after 48 h, we conducted our analyses at the 48 h post-treatment time point. In a parallel approach, the concentration of ML-SI1 was derived from our experimental data, and additional details elucidating this determination have been included in the Results section for clarity (Supplementary Figure 1). Additionally, the concentration of the seeded cells for analyses is noted in lines 117 – 118. The concentrations of cisplatin and ML-SI1 are shown in lines 130 - 131.

Reviewer 2 Report

Comments and Suggestions for Authors

In this work Kim and co-authors, showed the possible link between TRPML1-mediated lysosomal exocytosis and chemoresistance to cisplatin in ovarian cancer. They showed that the transporter is over expressed in chemo resistant cancer cell lines and its inibition leads to aninrease sensibility to cisplatin. Overall the work is well conducted and well written and discussion are well supported from the results. I suggest the publication of this manuscript after minor revisions. Here some comments:

- Check the entire manuscript for errors and typos, in particular I found an error in the tiutle, please check it.

-line 59-62: the word "lysosomes" is repeated many times and make sentences hard to be read.

-line 69-73: please add a reference to this sentence.

-line 75-80: I suggest to amplify the introduction with a better description of the channel and the associated problem regarding chemoresistance and the aim of the work.

- line 296: I suggest to explain the human KEGG pathway in the text.

Comments on the Quality of English Language

I suggest to check just for errors and typos. The work is well written.

Author Response

We would like to express our gratitude to the reviewers for taking time to assess the acceptability of our manuscript and for providing helpful suggestions. We carefully considered their concerns while revising the manuscript, and we believe that the revised version is an improved one. All changes have been highlighted in red in the revised manuscript. Our responses to the specific comments from the reviewers are provided below.

Comment 1. Check the entire manuscript for errors and typos, in particular I found an error in the tiutle, please check it.

Response 1: Thank you for the comment. We have fixed typos and errors in the manuscript.

Comment 2. line 59-62: the word "lysosomes" is repeated many times and make sentences hard to be read.

Response 2: We revised the description for better readability, as shown below (lines 66 – 68).

“The acidity of lysosomes creates an environment conducive to the accumulation of chemotherapeutic drugs with weak bases, resulting in their sequestration within these organelles [11,12]”.

Comment 3. line 69-73: please add a reference to this sentence.

Response 3. We have included references that were inadvertently omitted in the initial draft (line 93). The references are shown below.

  1. Asiago, V.M.; Alvarado, L.Z.; Shanaiah, N.; Gowda, G.A.; Owusu-Sarfo, K.; Ballas, R.A.; Raftery, D. Early detection of recurrent breast cancer using metabolite profiling. Cancer Res 2010, 70, 8309-8318, doi:10.1158/0008-5472.CAN-10-1319.
  2. Slupsky, C.M.; Steed, H.; Wells, T.H.; Dabbs, K.; Schepansky, A.; Capstick, V.; Faught, W.; Sawyer, M.B. Urine metabolite analysis offers potential early diagnosis of ovarian and breast cancers. Clin Cancer Res 2010, 16, 5835-5841, doi:10.1158/1078-0432.CCR-10-1434.
  3. Fong, M.Y.; McDunn, J.; Kakar, S.S. Identification of metabolites in the normal ovary and their transformation in primary and metastatic ovarian cancer. PLoS One 2011, 6, e19963, doi:10.1371/journal.pone.0019963.
  4. Fan, L.; Zhang, W.; Yin, M.; Zhang, T.; Wu, X.; Zhang, H.; Sun, M.; Li, Z.; Hou, Y.; Zhou, X.; et al. Identification of metabolic biomarkers to diagnose epithelial ovarian cancer using a UPLC/QTOF/MS platform. Acta Oncol 2012, 51, 473-479, doi:10.3109/0284186X.2011.648338.
  5. Zhang, T.; Wu, X.; Ke, C.; Yin, M.; Li, Z.; Fan, L.; Zhang, W.; Zhang, H.; Zhao, F.; Zhou, X.; et al. Identification of potential biomarkers for ovarian cancer by urinary metabolomic profiling. J Proteome Res 2013, 12, 505-512, doi:10.1021/pr3009572.

Comment 4. line 75-80: I suggest to amplify the introduction with a better description of the channel and the associated problem regarding chemoresistance and the aim of the work.

Response 4: TRPML1 plays a crucial role in the regulation of lysosomal trafficking, including exocytosis, achieved through the release of calcium from lumen into the cytosol. Our study suggests a significant involvement of lysosomal exocytosis in cisplatin sequestration, potentially contributing to the chemoresistance observed in ovarian cancer cells. Considering this, we believe that manipulating lysosomal exocytosis, specifically by targeting TRPML1, could present an innovative approach to overcoming chemoresistance in cancer treatment. Additional content has been integrated into the introduction for further clarification (lines 73 – 87).

Comment 5. line 296: I suggest to explain the human KEGG pathway in the text.

Response 5. We have added more explanation about the human KEGG pathway in the result section (lines 325 – 329).

“The top eight pathways emerged with low p-values are indicated in Figure 3D: 1) Alanine, aspartate, and glutamate metabolism, 2) Arginine and proline metabolism, 3) Aminoacyl-tRNA biosynthesis, 4) Purine metabolism, 5) Arginine biosynthesis, 6) Histidine metabolism, 7) Taurine and hypotaurine metabolism, and 8) Butanoate metabolism.”

Reviewer 3 Report

Comments and Suggestions for Authors

This is an interesting report on the possible mechanism of sensitivity of the classical anticancer drug cisplatin. The research is important for the understanding of the activity of platinum anticancer complexes. In this report, the link between the lysosomal calcium channel TRPML1 and cisplatin activity has been established, and the possibility of sensitizing ovarian cancer cells to cisplatin treatment has been shown. All experiments are well designed, and the paper can be accepted in its current form.

Author Response

We would like to express our gratitude to the reviewers for taking time to assess the acceptability of our manuscript and for providing helpful suggestions.

Thank you so much.